# REFERPIX2PIX: GUIDING MULTI-MODAL LLMS FOR IMAGE EDITING WITH REFERENTIAL PIXEL GROUNDING

## ABSTRACT

Instruction-based image editing methods allow user-friendly instruction to enhance controllability via natural command. However, without a user-provided mask, existing methods could not identify and edit specific objects if multiple similar instances exist, such as *"add the man on the right a hat"*. Furthermore, the iterative nature of the editing process may inherently involve ambiguous references from users, such as *'change it to blue'*, posing challenges in identifying the target without a contextual understanding. Multimodal large language models (MLLMs) offer impressive cross-modal comprehension and co-reference resolution capabilities. In this work, we present *ReferPix2Pix*, which leverages MLLMs to interpret editing instructions and provide regions of interest (RoI) for precise editing. Such pixel-grounded guidance from MLLMs enhances comprehension of referring expressions and resolves ambiguous references that facilitate localized editing of editing models. Additionally, we developed CoReferEdit benchmark to evaluate editing capabilities across iterative editing phases with multimodal co-references. Our comprehensive experiments show that our approach significantly enhances editing capability in referring and co-referential editing tasks. Our code and data will be made publicly available[1].

## 1 INTRODUCTION

As the need for visual content continues to grow across industries like photography, advertising, and social media, the role of image editing in improving and modifying images has become more crucial. Using natural language, an intuitive and adaptable tool, simplifies the guidance of the image editing process. Consequently, text-guided image editing has become increasingly favored, surpassing the popularity of other methods (Ling et al., 2021; Shi et al., 2022; Meng et al., 2021) that need users to specify editing regions.

Early text-based editing methods (Nam et al., 2018; El-Nouby et al., 2019; Meng et al., 2021; Hertz et al., 2022) relied on description-based captions, where the editing command outlines the desired image's attributes. This approach is not user-friendly, as it requires individuals to provide an extensive description of the target image rather than a straightforward editing instruction. InstPix2Pix (Brooks et al., 2023) is the first to collect a large-scale instruction-based editing dataset with input-goal-instruction triplet, where the instruction is generated by GPT-3, and the target image is synthesized from Prompt-to-Prompt (Hertz et al., 2022). MagicBrush (Zhang et al., 2024) introduces instruction-based interactive editing in the multi-round scenario and provides the edit mask annotations.

However, existing approaches have two notable limitations. First, they perform in benchmarks that contain images with a predominant single instance, which doesn't align with real-world scenarios where images often contain multiple instances. For instance, a user may want to specify an edit for one particular item, like *"change the shirt of **the right man** to blue"*. This requires the editing model to understand referring expressions. Unfortunately, current instruction-guided, mask-free methods fall short of accurately grounding these referring phrases, leading to incorrect edits that affect all instances in the image, not just the intended subject, as shown in fig. 3.

---

[1]Please refer to the anonymous webpage for code and qualitative results.

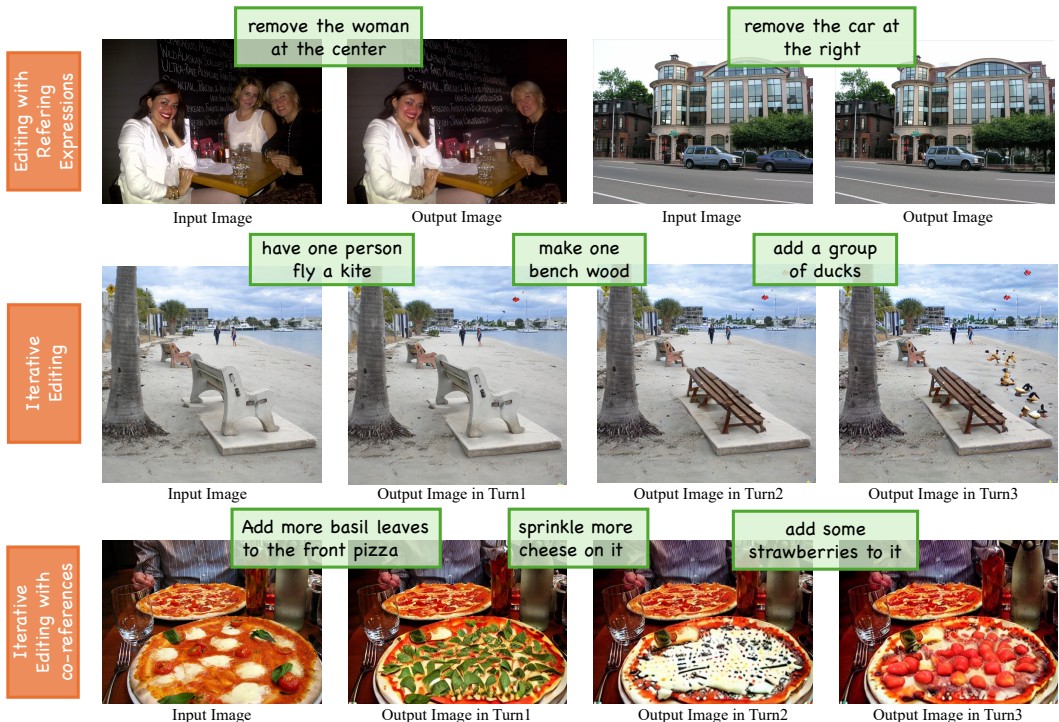

Figure 1: We introduce **ReferPix2Pix**, a novel approach that leverages MLLM's pixel-grounded guidance for advanced editing tasks. It demonstrates proficiency in (i) editing with referring expressions, (ii) multi-round iterative editing, and (iii) an innovative task we propose: iterative editing across multiple rounds incorporating multimodal co-references, designed to resonate with the intrinsic nature of user commands.

Moreover, the iterative nature of image editing introduces challenges with ambiguous co-references. For instance, after an initial instruction like *"change the shirt of **the right man** to blue"*, a subsequent command such as *"add **him** a hat"* can be unclear without a proper contextual understanding or memory of previous interaction. Although existing benchmark MagicBrush (Zhang et al., 2024) introduces multi-round editing with commands like *"have **him** a cowboy hat"* or *"wear **it** a necklace"*, however, there is only one dominant instance within the source image, thus not consider the scenario of ambiguous references in the editing conversation. Due to the absence of datasets with multimodal coreferences and the limitation of model design, current approaches struggle to resolve ambiguous references in multi-turn editing, as shown in fig. 4.

In this work, we harness the outstanding multi-modal compression capabilities of MLLMs to identify referring expressions and disambiguate references during editing sessions. Our approach leverages MLLM to direct a latent diffusion-based editing model, enabling precise localization of the target object without requiring explicit masks, as MLLM generates the intermediate editing mask. To tackle the data scarcity in referring edits, we adeptly modify the original ReferCOCO dataset (Kazemzadeh et al., 2014) for the referring editing task. In the first stage, the MLLM is trained to process interleaved source images and editing instructions. Its output is then mapped to the SAM-based model (Kirillov et al., 2023) to generate pixel-grounded guidance. In the second stage, we align the frozen MLLM and a diffusion-based editing model, where the MLLM's pixel-grounded guidance is used as conditional input of the editing model, ensuring referring/co-references editing.

Furthermore, to assess the model's ability in multi-modal co-reference resolution, we established a test set CoReferEdit by utilizing ReferCOCO (Kazemzadeh et al., 2014) annotations and GPT4V (OpenAI, 2023) generation, incorporating referring expressions in initial editing rounds and ambiguous references in the follow-up editing turns.

**Contributions.** Our contributions are summarized as follows:

- We introduce referring expression comprehension and multimodal co-reference resolution to interactive editing tasks to facilitate more natural editing instructions aligning with user commands in practice.

- We adapt MLLM by interlacing text and image inputs, empowering it to implicitly comprehend referring expressions and resolve ambiguous references, thus providing pixel-level guidance in interactive editing sessions.

- We establish the CoReferEdit benchmark to evaluate co-reference editing ability, complementing the limitation of the previous benchmark.

- Our model achieves superior performance in advanced image editing tasks with referring expressions and multimodal co-references.

## 2 RELATED WORK

**Text-based Image Editing.** *Description-based image editing:* Text-based editing models (Nam et al., 2018; El-Nouby et al., 2019) via GAN are limited to unrealistic synthesis. Diffusion models (Ho et al., 2020; Ramesh et al., 2022; Meng et al., 2021; Hertz et al., 2022), by controlling cross-modal attention maps between global description and latent pixels, achieve more semantically aligned manipulation. Local image editing enables detailed adjustments by filling in specified areas provided by users (Nichol et al., 2021; Couairon et al., 2022; Avrahami et al., 2022; Wang et al., 2023; Bar-Tal et al., 2022). *Instruction-guided image editing:* Different from description-based editing, instruction-guided editing (El-Nouby et al., 2019; Fu et al., 2020; Zhang et al., 2021) allows users to modify images by providing textual instructions, eliminating the need for detailed descriptions or region selection. InstPix2Pix (Brooks et al., 2023) constructs a large-scale instruction-based editing dataset by collecting synthetic texts from GPT-3 that finetuned on human-annotated instructions, and target images by (Hertz et al., 2022), enables image editing by following instructions. HIVE (Zhang et al., 2023b) utilizes training triplets and human ranking results to provide stronger supervision signals for better model training. MagicBrush (Zhang et al., 2024) introduces instruction-based interactive editing in the multi-round scenario. MGIE learns a projection from MLLMs to an editing model (Brooks et al., 2023) for instructional editing tasks. In this work, we advance the interactive editing task with referring expressions and co-reference resolution to facilitate more natural conversational editing in the real world.

**Referring Expression Comprehension.** Referring expression comprehension (REC) aims to localize a target object in an image described by a referring expression phrased in natural language. RefCOCO (Kazemzadeh et al., 2014) serves as valuable resources for tasks like referring expression segmentation, comprehension, and visual grounding. In this work, we introduce referring expressions to the image editing task, where the model is required to localize the edit object given an edit instruction with referring expressions.

**Multi-modal Reference Resolution.** Co-reference resolution is crucial in natural language processing (NLP), which involves identifying pronouns and the entities they refer to. Recent work (Seo et al., 2017) proposed visual co-reference resolution for Visual Question-Answering (VQA) dialogs, while (Rahman et al., 2023; Shen & Elhoseiny, 2023) extends visual co-reference to the story visualization setting. In this work, we investigate co-reference resolution within the context of interactive image editing tasks. It requires the model to identify and precisely modify the targeted object when users provide ambiguous references throughout multiple rounds of editing sessions.

**Multi-modal Large Language Models.** Large Language Models (LLMs) wield an extensive repository of human knowledge and exhibit impressive reasoning capabilities. Recent studies (Tsimpoukelli et al., 2021; Chen et al., 2022; Alayrac et al., 2022; Li et al., 2023b) utilize pre-trained language models to tackle vision-language tasks, and subsequent studies (Zhu et al., 2023; Zhang et al., 2023c; Li et al., 2023a; Huang et al., 2023; Chen et al., 2023) further enhance multi-modal abilities by aligning vision models with MLLMs input space. In addition to multi-modal comprehension, several works are dedicated to more challenging multi-modal generation tasks. Several current

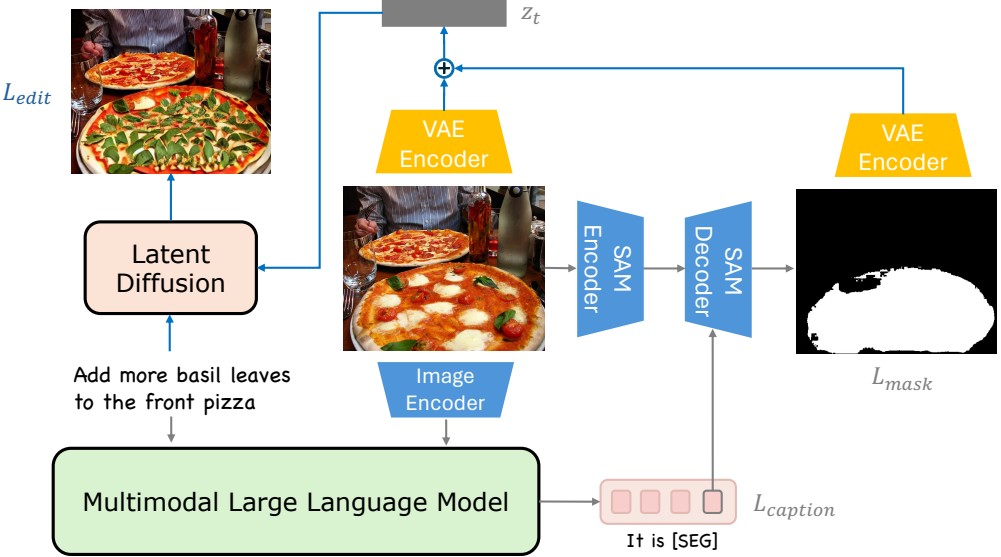

Figure 2: Model Pipeline. Gray arrows describe the first stage training with caption loss $\mathcal{L}_{caption}$ and mask loss $\mathcal{L}_{mask}$. Both gray and blue arrows show the pipeline of the second stage, where only the weight of latent diffusion is updated and calculated $\mathcal{L}_{edit}$, while other components remain frozen. We omit the forward diffusion step and VAE decoder for simplicity.

works (Koh et al., 2023; Wu et al., 2023; Zeqiang et al., 2023) learn a mapping from hidden embeddings of an LLM represents for additional visual outputs into the input space of a frozen pre-trained text-to-image generation model (Rombach et al., 2022). Similarly, MGIE (Fu et al., 2023) learns a projection from MLLMs to an editing model (Brooks et al., 2023) for instructional editing tasks. MLLMs can also excel in vision-centric tasks, such as object detection and segmentation (Rasheed et al., 2023; Lai et al., 2023; Wang et al., 2024; Zhang et al., 2023a). In this work, we leverage the exceptional reasoning and comprehension capabilities of MLLMs to offer guidance for advanced interactive editing tasks. Different MGIE that provides semantic guidance from MLLMs, which might lose fine-grained visual information, we leverage MLLM to provide pixel-grounded guidance for the editing model to effectively comprehend referring expressions and resolve ambiguous multimodal coreferences in multi-turn editing.

## 3 METHOD

MLLMs excel in vision-language tasks, such as image captioning (Li et al., 2023b) and grounding (Chen et al., 2023; Rasheed et al., 2023). MGIE is the first to use MLLMs to offer semantic guidance by mapping the hidden states of eight additional tokens onto a latent diffusion text conditioning space for image editing. However, such semantic-level guidance struggles to provide visual details for referring phrases within the editing instructions. In addition, the scarcity of multi-turn editing data with co-references hampers its performance on editing with co-references resolution.

To overcome the limitations of previous methods, we develop a two-stage pipeline for advanced editing tasks. In the first stage, the MLLM is trained to take images and editing instructions as input and produce pixel-level guidance. To circumvent data constraints, we innovatively repurpose a richly annotated image comprehension dataset for the referring editing task. In the second stage, we align a latent diffusion-based editing model with the first-stage MLLM, thereby enhancing its ability to comprehend referring expressions and resolve ambiguous co-references in multi-turn editing scenarios.

### 3.1 Generating Pixel-Grounded Guidance via MLLMs

In the first stage, we leverage MLLMs to comprehend interleaved images and textual instructions, thereby facilitating pixel-level grounding that enhances precise editing guidance. More specifically, the input image $x_{img}$ is encoded by an image encoder CLIP ViT-H/14 (Radford et al., 2021), and its visual representation $I_x$ is projected to the input space of LLM $\mathcal{L}$ denoted as $f_{v2t}(I_x)$. To facilitate fine-grained pixel-level object grounding, similar to (Rasheed et al., 2023), we utilize a pretrained SAM (Kirillov et al., 2023) encoder as a segmentation image encoder $\mathcal{E}_{SAM}$ and a SAM-based segmentation decoder $\mathcal{D}_{SAM}$. We add a new token [SEG] to the LLM vocabulary, and the last hidden state $h_{\text{[SEG]}}$ of [SEG] token is projected to the segmentation decoder's input space, denoted as $f_{t2m}(h_{\text{[SEG]}})$. Therefore, the MLLM is trained to predict the ground truth text, represented as $y_t = \mathcal{L}(f_{v2t}(I_x), x_t)$. Its last hidden state of [SEG], as well as the encoded feature by SAM encoder $\mathcal{E}_{SAM}(x_{img})$ are taken as input to the decoder $\mathcal{D}_{SAM}$ to produce segmentation mask $M$, defined as follows:

$$M = \mathcal{D}_{SAM}(f_{t2m}(h_{\text{[SEG]}}), \mathcal{E}_{SAM}(x_{img})) \tag{1}$$

The predicted output $y_t$ and the segmentation mask $M$ are used for calculating the caption loss $\mathcal{L}_{caption}$ and the mask loss $\mathcal{L}_{mask}$ respectively. The whole pipeline of the first stage is demonstrated in fig. 2 with gray arrows.

#### 3.1.1 Training data and prompts design.

For datasets with input-goal-instruction triplet, along with the editing masks, e.g., MagicBrush (Zhang et al., 2024), we can directly use it as our instruction finetuning data. The input prompt is defined as follows:

***User:*** *The <image> provides an overview of the picture. Given this editing instruction: {edit instruction}. Please segment the edited region in this image.*

***Assistant:*** *Sure, it is [SEG].*

where the <image> token is replaced by 256 tokens generated by the image encoder $\mathcal{E}$. The MLLM learns to produce *"Sure, it is [SEG]"*, and the last hidden state of the [SEG] is then passed through the segmentation decoder $\mathcal{D}$ to produce segmentation mask as mentioned above.

**Editing with Referring Expressions.** However, there is no large-scale editing dataset with editing mask annotations. Therefore, we have devised a strategy to effectively leverage annotated data from other tasks, repurposing them for editing instruction training. Specifically, we turn our attention to ReferCOCO (Kazemzadeh et al., 2014), a dataset originally curated for referring expression comprehension, which features multiple object instances within each image. Each of these instances is annotated with both referring expressions and corresponding segmentation masks By harnessing this richly annotated data, we can ingeniously adapt ReferCOCO (Kazemzadeh et al., 2014) for our edit instruction training needs through the automated generation of a comprehensive set of edit instructions derived from the dataset's existing annotations. The modified version is denoted as ReferCOCO$_{edit}$. Below is an example in the automatically generated list of editing templates: *"replace {class name} with {new class}"*, where {*class name*} represents the referring expression corresponding to an instance within a ReferCOCO image, while {*new class*} is obtained by randomly sampling from the set of COCO object classes. Please refer to the supplementary to find the full list of automatically generated edit templates.

This design enables us to adeptly harness the referring expressions within ReferCOCO to generate edit instructions with referring phrases and utilize the corresponding segmentation masks as editing guides. This eliminates the need for manual annotation, editing of masks, or the generation of target images for training, thereby effectively circumventing the constraints posed by the scarcity of instruction-based datasets with mask annotations.

**Editing with Multimodal Coreferences.** Furthermore, we utilize a similar design to construct the training data, which enables the model to understand multi-modal co-references within an iterative editing session, denoted as ReferCOCO$_{edit}^{coref}$. The first round of edit instructions adheres to the procedure mentioned above. In follow-up turns, the instructions are deliberately revised to incorporate

an ambiguous reference, thus training the model to adeptly resolve ambiguous multimodal references within an editing session. For instance: *"add {reference} {new class}"*, where *{reference}* represents for *"he, she, they, it"* based on different scenarios, and *{new class}* is randomly sampled from COCO object class. Below is one training example:

***User:*** *The <image> provides an overview of the picture. Given this editing instruction: give the right man a pair of glasses. Please segment the edited region in this image.*

***Assistant:*** *Sure, it is [SEG].*

***User:*** *Given this editing instruction: add him a hat. Please segment the edited region in this image.*

***Assistant:*** *Sure, it is [SEG].*

Our training methodology effectively links the ambiguous references *'him'* with the contextually mentioned *"the right man"* as well as the corresponding visual features within the image. By minimizing the $\mathcal{L}_{mask}$ loss between the predicted masks of corresponding `[SEG]` tokens and the ground truth masks, the MLLM learns to provide pixel-grounded guidance given the edit instructions and multimodal context.

## 3.2 MLLM GUIDANCE FOR REFERRING/CO-REFERENCES EDITING

During the second stage, we freeze the first stage MLLM and train a diffusion model conditioned on the input source image $x^s_{img}$, editing guidance provided by the MLLM $M$, and editing instruction $x_t$. We build on top of latent diffusion (Rombach et al., 2022) that learns to generate data samples through a sequence of denoising in the latent space of a pretrained variational autoencoder with encoder $\mathcal{E}_{VAE}$ and decoder $\mathcal{D}_{VAE}$. More specifically, as shown in fig. 2, the MLLM will take as input the source image $x^s_{img}$ as well as the edit instruction $x_t$ to produce pixel-grounded guidance. For an input target image $x^t_{img}$, the diffusion process adds noise to the encoded latent $z = \mathcal{E}_{VAE}(x^t_{img})$, producing a noisy latent $z_t$ where the noise level increases over timesteps $t \in T$. Then we channel-wise concatenate the encoded source image feature $\mathcal{E}_{VAE}(x^s_{img})$ and pixel-grounded guidance $M$ from MLLM in eq. (1) as image condition, defined as follows:

$$c_I = \texttt{concat}(\mathcal{E}_{VAE}(x^s_{img}), M) \tag{2}$$

We add input channels to the first convolutional layer to support image and mask conditioning by concatenating $z_t$ and $c_I$. The cross-attention condition $c_T$ is the edit instruction $x_t$ encoded by the text encoder. The editing loss is calculated as follows:

$$\begin{aligned}
\tilde{e}_\theta(z_t, c_I, c_T) &= e_\theta(z_t, \varnothing, \varnothing) \\
&+ \alpha_I \cdot (e_\theta(z_t, c_I, \varnothing) - e_\theta(z_t, \varnothing, \varnothing)) \\
&+ \alpha_T \cdot (e_\theta(z_t, c_I, c_T) - e_\theta(z_t, c_I, \varnothing)) \\
\mathcal{L}_{\text{edit}} &= \mathbb{E}_{z, c_I, c_T, \epsilon \sim \mathcal{N}(0,1), t} \left[ ||\epsilon - \epsilon_\theta(z_t, t, c_I, c_T)||^2_2 \right]
\end{aligned} \tag{3}$$

where $\alpha_I$ and $\alpha_T$ are the weights of the guidance scale for the image and the instruction. We randomly set $c_I = \varnothing$, $c_T = \varnothing$, or both $c_I = \varnothing$ and $c_T = \varnothing$ for 5% of data during training for classifier-free guidance similar to InstPix2Pix (Brooks et al., 2023).

The second-stage end-to-end training design that uses the mask predicted by MLLMs as the conditioned input for latent diffusion, rather than solely depending on the ground truth mask during training, is anchored in a critical insight: the inherent discrepancy between the ground-truth mask and the predicted mask. Utilizing the ground truth mask as the sole training input could lead to a scenario where, during inference, the editing model might indiscriminately modify every region indicated by the MLLM's mask, resulting in suboptimal editing outcomes, especially if the guidance from MLLM lacks precision. To mitigate this risk and enhance the model's performance, we mix up the ground truth masks and predicted masks by MLLM as the conditional input for latent diffusion, thereby ensuring greater flexibility and robustness in its editing capabilities.

## 4 CoReferEdit Data Collection.

Although MagicBrush (Zhang et al., 2024) includes references in the multi-round editing session, such as *"Have him a cowboy hat"* or *"Wear it a necklace"*. However, there is only one object instance in the image, thereby it does not consider the scenario of multiple instances in one image, where the model is required to identify the target edit object across different instances given an ambiguous reference. Hence, we design an automatic pipeline to collect a test set to evaluate the model's editing ability in a multi-round co-reference resolution setting.

Specifically, we consider ReferCOCO (Kazemzadeh et al., 2014) images since they contain multiple instances in an image and the corresponding referring expressions. Then we fed randomly sampled image, the edit object with the referring expression, such as *"the man on the right"* and original caption to `gpt-4-vision-preview` (OpenAI, 2023). In the first round, GPT4V is used to generate an edit instruction regarding the input edit object, a global caption that modifies the original caption based on the generated edit instructions, and a local caption that focuses on describing the edit object only. In the follow-up turns, GPT4V is prompted to generate edit instructions regarding the same object but uses ambiguous references for the edit object and generates a new global/local caption. fig. 10 demonstrates an example in our collected CoReferEdit data, and fig. 9 shows the distribution of edit object class and edit instruction in the collected set. All the edit sessions include 3 rounds of editing, and after manual quality control, there are 403 editing sessions and 1196 edit turns in the collected test set.

## 5 Experiments

### 5.1 Datasets and Evaluation metrics

**MagicBrush.** MagicBrush (Zhang et al., 2024) features multiple rounds of editing within its sessions, with 8,807 editing turns for training, 528 for dev, and 1,053 for test set. We adhere to the original train/dev/test splits for training and evaluating models. The results are in the context of multi-round editing, where the images edited in the final turn are evaluated using L1/L2 distance, CLIP (Radford et al., 2021) image-image similarity (CLIP-I), CLIP (Radford et al., 2021) image-text similarity (CLIP-T), and the DINO (Zhang et al., 2022) score, consistent with MagicBrush (Zhang et al., 2024). We report the final turn results to evaluate the editing capability in multi-round editing.

**GQA-Inpaint.** GQA-Inpaint (Yildirim et al., 2023) was built on top of the GQA Dataset (Hudson & Manning, 2019) which includes multiple instances and referring expressions for the images. GQA-Inpaint leverages the annotations in GQA and designed editing instructions containing referring expressions such as *"remove the woman at the right of the boat"*, where the edit object is selected from the scene graphs of GQA. All the comparison methods report zero-shot performance using L1/L2 distance, CLIP (Radford et al., 2021) image similarity (CLIP-I), and DINO (Zhang et al., 2022) score on this dataset. We utilize this dataset to assess the editing capability with referring expressions.

**CoReferEdit.** We mentioned the collection pipeline and data distribution details in section 4. The test set contains 403 edit sessions and 1196 edit turns. We facilitate the evaluation of the collected dataset by calculating the global or local CLIP text-image similarity for the final turn, or all turns on average. For local caption, the edited image is cropped based on the bounding box of the edit object to calculate the local image-text similarity. We utilize this dataset to assess the editing capability with ambiguous references in multi-round editing.

### 5.2 Comparison Approaches.

We compare our method with the state-of-the-art instruction-based editing approaches: HIVE (Zhang et al., 2023b), InstPix2Pix (Brooks et al., 2023), MGIE (Fu et al., 2023). Inst-Pix2Pix (Brooks et al., 2023) take the concatenation of encoded source image and latent noise vector as input to latent diffusion model and conditioned on edit instruction to produce the target image. HIVE (Zhang et al., 2023b) relies on human feedback on edited images to learn what users generally prefer and uses this information to fine-tune InstPix2Pix (Brooks et al., 2023), aiming to align

| Method | GQA Inpaint | | | | CoReferEdit (Final) | | CoReferEdit (All) | |
|---|---|---|---|---|---|---|---|---|
| | L1↓ | L2↓ | CLIP-I↑ | DINO↑ | Local | Global | Local | Global |
| HIVE (Zhang et al., 2023b) | 0.1051 | 0.0326 | 0.8379 | 0.7296 | 0.2574 | 0.3075 | 0.2489 | 0.3014 |
| InstPix2Pix (Brooks et al., 2023) | 0.1182 | 0.0364 | 0.792 | 0.6435 | 0.2547 | 0.3106 | 0.2617 | 0.3132 |
| MGIE (Fu et al., 2023) | 0.0916 | 0.0328 | 0.8728 | 0.7819 | 0.2507 | 0.3073 | 0.2541 | 0.3065 |
| ReferPix2Pix (ours) | **0.0822** | **0.0231** | **0.9020** | **0.8551** | **0.2643** | **0.3187** | **0.2732** | **0.3204** |

Table 1: Left: Zero-shot performance on GQA Inpaint, which contains editing instructions with referring expressions. Right: Zero-shot performance on our CoReferEdit dataset.

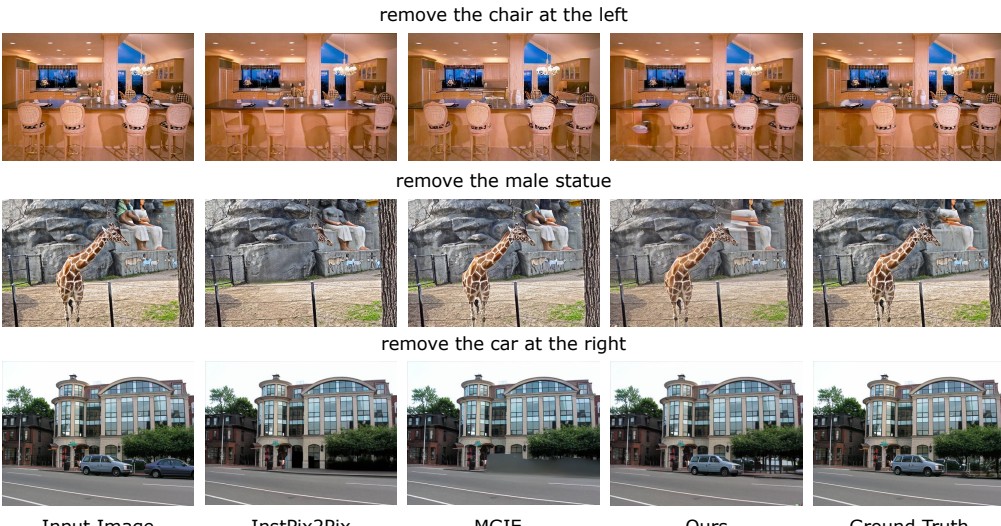

remove the chair at the left

remove the male statue

remove the car at the right

| Input Image | InstPix2Pix | MGIE | Ours | Ground Truth |

Figure 3: Qualitative result on GQA Inpaint (Yildirim et al., 2023), which contains single-turn editing instruction with referring expressions.

more closely with human expectations. MGIE (Fu et al., 2023) leverages MLLMs to produce visual imagination as explicit semantic guidance for the editing model.

### 5.3 IMPLEMENTATION DETAILS.

In the first stage, we use MagicBrush (Zhang et al., 2024), ReferCOCO$_{edit}$ and ReferCOCO$_{edit}^{coref}$ as the training data. The MLLM is trained with captioning loss, Mask BCELoss, and Mask DICELoss. The training batch size is 16 and uses AdamW optimizer with learning rate $1e - 4$ for 4 epochs. We use MagicBrush (Zhang et al., 2024) and modified ReferCOCO (Kazemzadeh et al., 2014) for the first stage of training. In the second stage, the first stage model is kept frozen, and we only train the Unet of the latent diffusion. The input channel of the first convolution layer is set to 12. The training is conducted with a batch size of 64 and a learning rate of $1e - 4$ over $4k$ steps. We use MagicBrush (Zhang et al., 2024) and InstPix2Pix (Brooks et al., 2023) as the training data in the second stage. $\alpha_I$ and $\alpha_T$ in eq. (3) are set to be 1.5 and 7.5 respectively. All experiments are conducted in PyTorch on 2 80G A100 GPUs.

### 5.4 EXPERIMENTAL RESULTS

#### 5.4.1 EDITING WITH REFERRING EXPRESSIONS

We choose GQA-Inpaint (Yildirim et al., 2023) to evaluate the editing ability with referring expressions and report the quantitative result in table 1 (left). Our approach outperforms all the baseline models, illustrating that our model excels at recognizing referring expressions and precisely editing the corresponding object.

fig. 3 showcases the qualitative results on the GQA Inpaint dataset. The baseline models struggle to localize the target region given referring expressions. Take the first row as an example, Inst-

| Method | MagicBrush | | | | |
|--------|------|------|---------|-------|---------|
| | L1↓ | L2↓ | CLIP-I↑ | DINO↑ | CLIP-T↑ |
| HIVE (Zhang et al., 2023b) | 0.0966 | 0.0365 | 0.8785 | 0.7891 | **0.2796** |
| InstPix2Pix (Brooks et al., 2023) | 0.0964 | 0.0353 | 0.8924 | **0.8273** | 0.2754 |
| MGIE (Fu et al., 2023) | 0.1208 | 0.0507 | 0.8582 | 0.7559 | 0.2772 |
| ReferPix2Pix (ours) | **0.0885** | **0.0297** | **0.8987** | 0.8182 | 0.2783 |
| ReferPix2Pix (w/o comb) | 0.0911 | 0.0309 | 0.8870 | 0.8081 | 0.2732 |
| ReferPix2Pix w/ GT mask (upper bound) | 0.0762 | 0.0245 | 0.9145 | 0.8682 | 0.2792 |
| GT image | - | - | - | - | 0.2829 |

Table 2: Quantitative result on MagicBrush (Zhang et al., 2024). All the models are trained on both MagicBrush (Zhang et al., 2024) and InstPix2Pix (Fu et al., 2023). The best-performing results are highlighted in bold, while the second-best are underlined. w/o comb indicates that the editing model is trained independently without integrating the MLLM and instead takes ground truth masks during training. w/ GT mask means the ending model takes ground truth masks as input during inference serving as a upper bound.

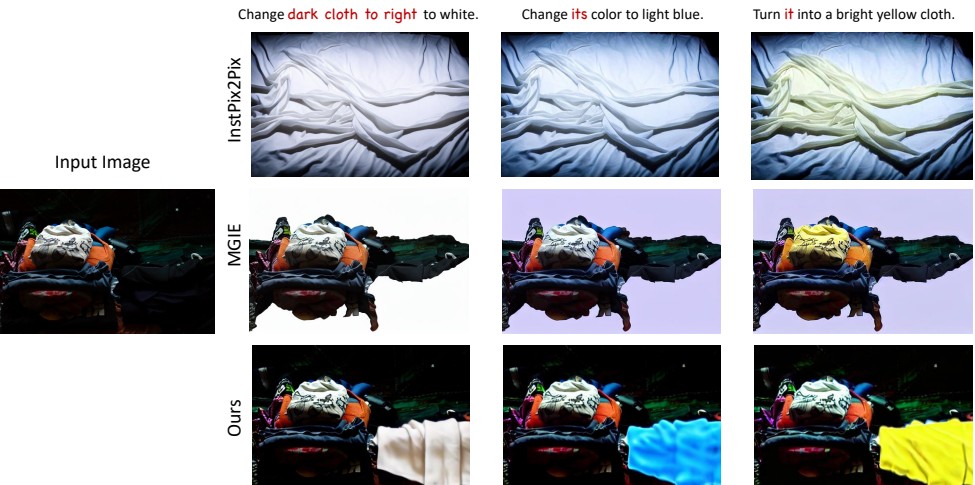

Figure 4: Qualitative result on CoReferEdit.

Pix2Pix (Brooks et al., 2023) appears to remove the mats of the left chairs, whereas MGIE removes the items on the table rather than the left chair. In contrast, our model identifies the region indicated by the referring phrases and performs the appropriate edits.

### 5.4.2 EDITING IN MULTI-TURNS

We utilize MagicBrush (Zhang et al., 2024) to evaluate the editing capabilities of the model in an iterative context. The scores for the final round are presented in table 2 (upper). Our model achieves better L1/L2 and CLIP-I scores while reaching comparable results in DINO and CLIP-T metrics. The reason our model doesn't significantly surpass other models is due to our method enhancements in referential editing. However, the images in MagicBrush (Zhang et al., 2024) predominantly feature a single object, which does not necessitate the capability to distinguish among multiple instances. In addition, the CLIP-T shows minimal differentiation between methods, with the ground truth image-text similarity being only 2.65% higher than that of the lowest-performing model.

### 5.4.3 EDITING IN MULTI-TURNS WITH CO-REFERENCES

We use CoReferEdit to evaluate the model's capability in multi-round editing involving multimodal co-references. table 1 (right) shows the model's performance, evaluated using CLIP text-image similarity based on local/global descriptions across all/final rounds. Owing to the MLLM's capabil-

| Method | GQA Inpaint | | | | CoReferEdit (Final) | | CoReferEdit (All) | |
|---|---|---|---|---|---|---|---|---|
| | L1↓ | L2↓ | CLIP-I↑ | DINO↑ | Local | Global | Local | Global |
| w/o co-refer | 0.0824 | 0.0232 | 0.9014 | 0.8546 | 0.2581 | 0.3119 | 0.2620 | 0.3154 |
| w/o comb | 0.0821 | 0.0231 | 0.9015 | 0.8554 | 0.2603 | 0.3147 | 0.2691 | 0.3184 |
| Ours (default) | 0.0822 | 0.0231 | 0.9020 | 0.8551 | 0.2643 | 0.3187 | 0.2732 | 0.3204 |

Table 3: Ablation study on GQA Inpaint (Yildirim et al., 2023) and CoReferEdit. w/o co-refer represents for without the multimodal coreferences data mentioned in section 3.1.1. w/o comb indicates that the editing model is trained independently without integrating the MLLM and instead takes ground truth masks during training.

ity in understanding contextual data and deciphering multimodal ambiguous references, our model achieves superior performance, particularly in local similarity, compared to other models. Please refer to the supplementary for human evaluation results.

fig. 4 demonstrates the qualitative results on CoReferEdit, starting with the initial round of editing using reference phrases, followed by edits involving ambiguous references. InstPix2Pix (Brooks et al., 2023) tends to modify the entire image. MGIE (Fu et al., 2023) struggles to identify the *"dark cloth to the right"* and thus turns all black areas to white. Furthermore, in the final round, it fails to recognize the ambiguous referring word *'it'* and mistakenly alters the cloth in the center to yellow. In contrast, our method precisely identifies the target object in the first round and iteratively edits the correct object by associating the ambiguous reference *'it'* with both the contextually mentioned *"dark cloth to the right"* and the corresponding visual pixels in the image.

## 5.5 ABLATION STUDY

table 3 shows the effect of coreference training and end-to-end combined training, where w/o co-refer represents without the multimodal coreferences data mentioned in section 3.1.1, and w/o comb indicates that the editing model is trained independently without integrating the MLLM and instead takes ground truth masks during training. The co-refer training did not affect performance on GQA Inpaint since there are no ambiguous references in the dataset. However, it enhanced the performance on CoReferEdit by a large margin in the multimodal coreference editing scenario.

The combined training improves performance on all three datasets, i.e., GQA Inpaint (Yildirim et al., 2023) and CoReferEdit in table 3, as well as MagicBrush (Zhang et al., 2024) in table 2 (bottom). This is because separately training the latent diffusion with the ground truth mask as the input could lead to a scenario where, during inference, the editing model might indiscriminately modify every region indicated by the MLLM's mask, resulting in suboptimal editing outcomes. Additionally, table 2 presents results using a ground truth mask (w/ GT mask) as the editing model input, serving as an upper bound. Enhanced editing performance with an accurate mask offers practical application potential, especially when users can adjust the mask if the MLLM-generated one is suboptimal.

## 6 CONCLUSION

In conclusion, we first discussed the limitations of existing instruction-based image editing methods that struggle with identifying and modifying specific objects in the presence of multiple instances without user-provided masks. The challenge is further compounded during iterative editing processes, where vague references like *'change it to blue'* require a contextual understanding to accurately identify the target. We introduce *ReferPix2Pix*, which utilizes the MLLM's multimodal reasoning comprehension and co-reference resolution capabilities for advanced editing tasks. This enables the interpretation of editing instructions and the provision of precise RoI for image editing, thereby significantly improving the ability to understand referring expressions and resolve ambiguous references in iterative editing turns. Furthermore, we established CoReferEdit for evaluating the performance of editing models in handling co-referential editing tasks. Our comprehensive experiments show that our approach significantly enhances editing capability in referring and co-referential editing tasks.

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

## A  ABLATION STUDY

table 4 shows the analysis of the impact of training data on the accuracy of mask prediction, which subsequently influences the editing performance on MagicBrush (Zhang et al., 2024) test split. gIoU is defined by the average of all per-image Intersection-over-Unions (IoUs), while cIoU is defined by the cumulative intersection over the cumulative union. The mask Recall metric computes the IoU between the predicted masks and the ground truth editing mask. Predictions with an IoU greater than the threshold of 0.5 are considered in the count.

When we remove the $\text{ReferCOCO}_{edit}^{ref}$ training data, there is not a significant decrease in performance. This is because MagicBrush (Zhang et al., 2024) does not have ambiguous references in multi-turn editing. However, removing the $\text{ReferCOCO}_{edit}$ training data in advance leads to a substantial drop in mask prediction accuracy (gIoU/cIoU /Recall). This is due to the lack of large-scale training data that provides precise masks for the editing regions, consequently deteriorating the editing performance (the remaining metrics).

| Method | gIoU | cIoU | Recall | L1↓ | L2↓ | CLIP-I↑ | DINO↑ | CLIP-T↑ |
|---|---|---|---|---|---|---|---|---|
| Ours (default) | 0.3018 | 0.3292 | 0.9602 | 0.0885 | 0.0297 | 0.8987 | 0.8182 | 0.2783 |
| - ReferCOCO$^{ref}_{edit}$ | 0.2910 | 0.3215 | 0.9545 | 0.0868 | 0.0292 | 0.8921 | 0.8290 | 0.2732 |
| - ReferCOCO$_{edit}$ | 0.2638 | 0.2783 | 0.9356 | 0.0902 | 0.0306 | 0.8882 | 0.8103 | 0.2695 |

Table 4: An analysis of the impact of training data on the accuracy of mask prediction, which subsequently influences the editing performance on MagicBrush (Zhang et al., 2024).

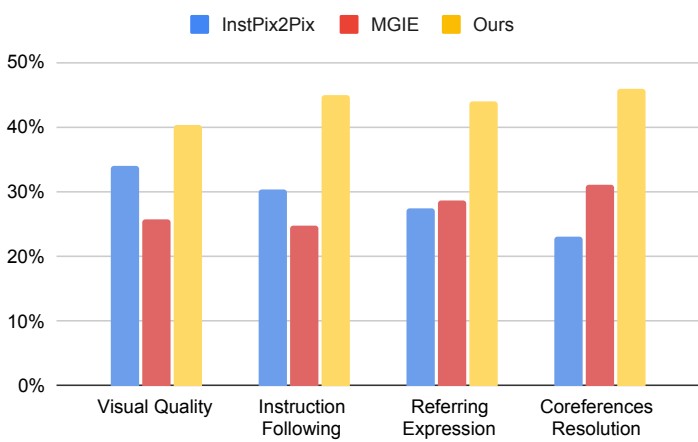

Figure 5: Human Evaluation Results on CoReferEdit.

**Comparison of using only the SAM-generated mask as a visual condition.** SAM-generated masks involve using LLM+SAM since SAM alone cannot accept edit instructions. We used the LLM aligned with the SAM decoder to produce segmentation masks without fine-tuning for editing instructions. The results indicate that in a zero-shot setting, the in-context editing instructions for LLM+SAM mask generation cannot accurately identify precise editing regions, as shown in the appendix A.

| Method | GQA Inpaint | | | | CoReferEdit (Final) | |
|---|---|---|---|---|---|---|
| | L1↓ | L2↓ | CLIP-I↑ | DINO↑ | Local | Global |
| LLM+SAM mask | 0.0912 | 0.0314 | 0.8872 | 0.8435 | 0.2493 | 0.3041 |
| Ours (default) | 0.0822 | 0.0231 | 0.9020 | 0.8551 | 0.2643 | 0.3187 |

## B HUMAN EVALUATION

In addition, we use Mechanical Turk to assess the quality of 100 editing sessions produced by our methods or baselines InstPix2Pix (Brooks et al., 2023) and MGIE (Fu et al., 2023) on CoReferEdit. MTurkers are tasked with evaluating pairs of editing instructions and the corresponding edited images to determine which model excels in terms of visual quality, adherence to the editing instructions, the ability of referring expression comprehension (REC), and ambiguous reference resolution. Each pair is evaluated by 3 unique workers. We evaluate the REG ability by asking MTurkers to assess the first-turn editing result, while final-turn for coreference resolution. The results presented in fig. 5 demonstrate that our model, enhanced with outstanding multimodal comprehension capabilities and directed by pixel-based editing guidance, achieves superior editing performance in all aspects.

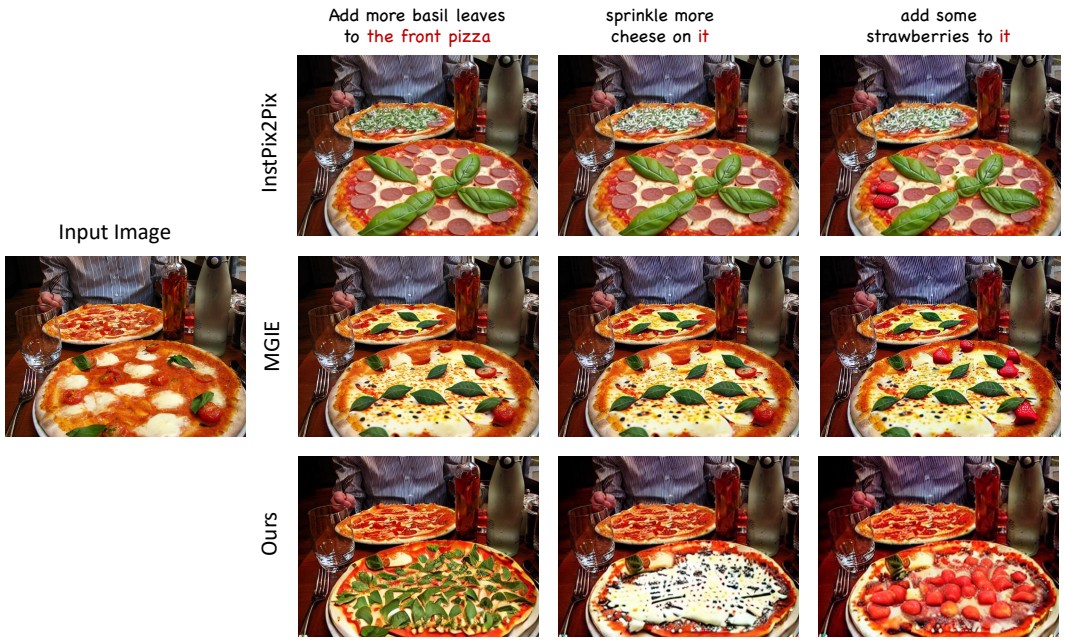

Figure 6: Qualitative results on CoreferEdit.

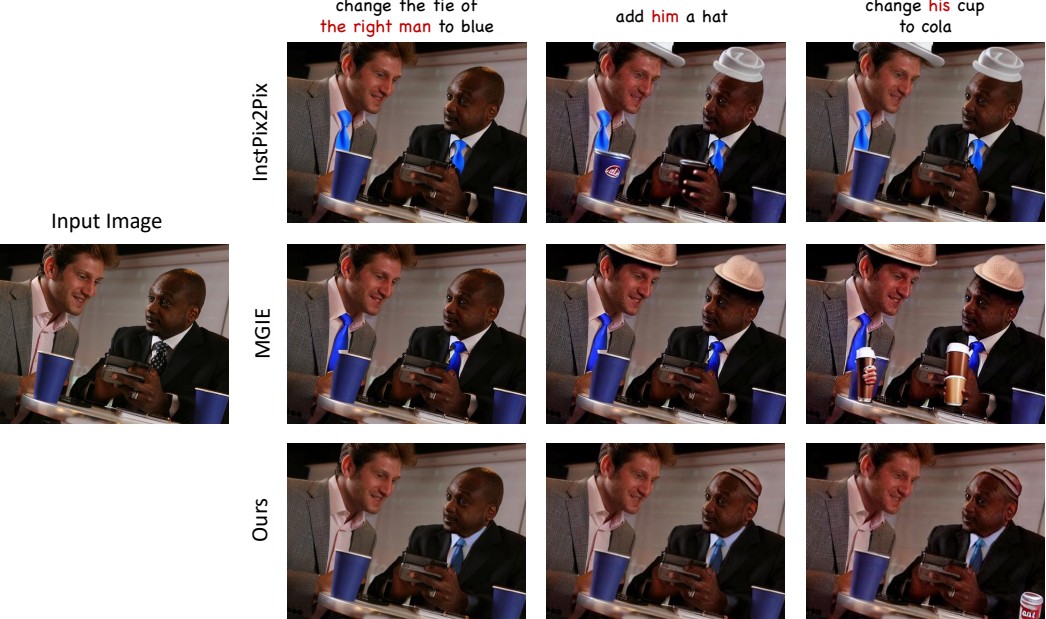

Figure 7: Qualitative results on CoreferEdit.

## C QUALITATIVE RESULTS

figs. 6 to 8 show qualitative comparison between InstPix2Pix (Brooks et al., 2023), MGIE (Fu et al., 2023) and our method. For the first editing turn in the three examples, InstPix2Pix (Brooks et al., 2023) and MGIE (Fu et al., 2023) struggle to identify the referring expressions, e.g., *"the right person"* in fig. 8, and change hats of both people to red color. In the following turns, they iteratively alter the color of both jackets and fail to resolve the reference word *"his"* in the editing instructions.

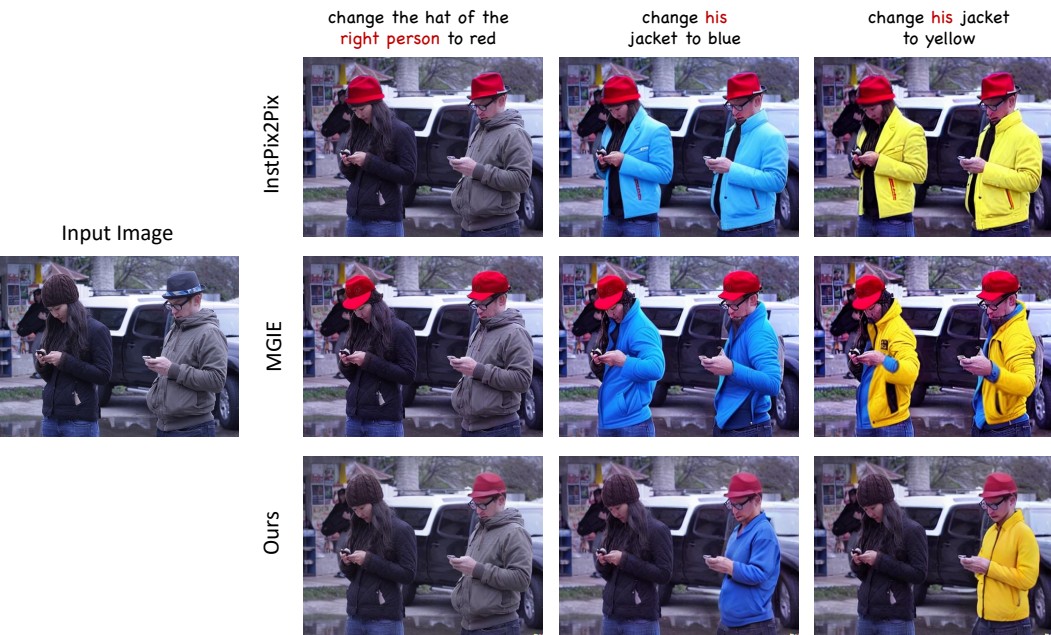

Figure 8: Qualitative results on CoreferEdit.

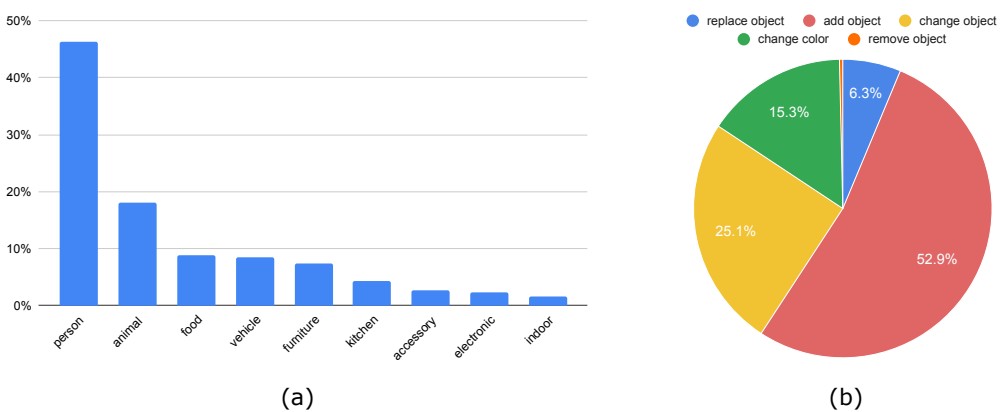

Figure 9: The data distribution of our collected CoReferEdit (a) Edit object class distribution following COCO object supercategory. (b) Edit instruction type distribution.

# D EDITING INSTRUCTION PROMPTS

Below, we show our designed editing instruction prompts for MagicBrush (Zhang et al., 2024), and our ReferCOCO$_{edit}$ and ReferCOCO$_{edit}^{ref}$ adapted based on ReferCOCO (Kazemzadeh et al., 2014) during the training stage.

## D.1 EDIT INSTRUCTION TEMPLATES FOR MAGICBRUSH (ZHANG ET AL., 2024)

*"Can you segment the region that should be edited in this image?"*

*"Please segment the edited region in this image."*

*"What region should be edited in this image? Please respond with a segmentation mask."*

*"What is the edited region in this image? Please output segmentation mask."*

**Input Image**

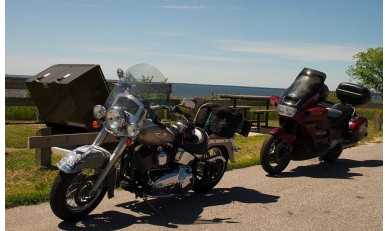

**Edit object**
cycle in back

**Bounding box**
[429.66, 112.36, 196.86, 181.57]

- - - - - - - - - - - - - - - - - - - - - - - - - - - - - - - - - - - - - - - - - - - -

**Edit Session**

{
"edit_instruction": "Add white stripes to the cycle in back.",
"global_caption": "Two motorcycles with the back one featuring white stripes are parked next to a park grill.",
"local_caption": "Motorcycle with white stripes."
},
{
"edit_instruction": "Add a red cover to its seat.",
"global_caption": "Two motorcycles with the back one featuring white stripes and a red seat cover are parked next to a park grill.",
"local_caption": "Motorcycle with white stripes and red seat cover."
},
{
"edit_instruction": "Give it white wall tires.",
"global_caption": "Two motorcycles with the back one featuring white stripes, white wall tires and a red seat cover are parked next to a park grill.",
"local_caption": "Motorcycle with white stripes, red seat cover and white wall tires."
}

Figure 10: An example in our CoReferEdit benchmark.

"Could you provide a segmentation mask for the edited region in this image?"

"Please identify and segment the edited region in this image."

"Where is the region should be edited in this picture? Please respond with a segmentation mask."

"Can you highlight the region that should be edited in this image with a segmentation mask?"

### D.2 EDIT INSTRUCTION TEMPLATES FOR REFERCOCO$_{edit}$

"Given this edit instruction: change {class_name} to {color}. Can you segment the region that should be edited in this image?"

"Given this edit instruction: add {new_class} on {class_name}. Please segment the edited region in this image."

"Given this edit instruction: make {class_name} {color}. What region should be edited in this image? Please respond with a segmentation mask."

"Given this edit instruction: replace {class_name} with {new_class}. What is the edited region in this image? Please output segmentation mask."

"Given this edit instruction: remove {class_name}. Could you provide a segmentation mask for the edited region in this image?"

"Given this edit instruction: put {new_class} on {class_name}. Please identify and segment the edited region in this image."

*"Given this edit instruction: let {class_name} be {new_class}. Where should the region be edited in this picture? Please respond with a segmentation mask."*

*"Given this edit instruction: make {class_name} be {shape}. Can you highlight the region that should be edited in this image with a segmentation mask?"*

where {*class_name*} represents the referring expression corresponding to an instance within a Refer-COCO image, while {*new_class*} is obtained by randomly sampling from the set of COCO object classes.

### D.3   EDIT INSTRUCTION TEMPLATES FOR REFERCOCO$_{edit}^{ref}$

*"Given this edit instruction: change {reference} to {color}. Can you segment the region that should be edited in this image?"*

*"Given this edit instruction: add {new_class} on {reference}. Please segment the edited region in this image."*

*"Given this edit instruction: make {reference} {color}. What region should be edited in this image? Please respond with a segmentation mask."*

*"Given this edit instruction: replace {reference} with {new_class}. What is the edited region in this image? Please output segmentation mask."*

*"Given this edit instruction: remove {reference}. Could you provide a segmentation mask for the edited region in this image?"*

*"Given this edit instruction: put {new_class} on {reference}. Please identify and segment the edited region in this image."*

*"Given this edit instruction: let {reference} be {new_class}. Where should the region be edited in this picture? Please respond with a segmentation mask."*

*"Given this edit instruction: make {reference} be {shape}. Can you highlight the region that should be edited in this image with a segmentation mask?"*

where {*reference*} represents for *"he, she, they, it"* based on different scenarios, and {*new_class*} is randomly sampled from COCO object class.

## E   IMPLEMENTATION DETAILS.

In the first stage, we use MagicBrush (Zhang et al., 2024), ReferCOCO$_{edit}$ and ReferCOCO$_{edit}^{coref}$ as the training data. The MLLM is trained with captioning loss, Mask BCELoss, and Mask DICELoss. The training batch size is 16 and uses AdamW optimizer with learning rate $1e-4$ for 4 epochs. We use MagicBrush (Zhang et al., 2024) and modified ReferCOCO (Kazemzadeh et al., 2014) for the first stage of training. In the second stage, the first stage model is kept frozen, and we only train the Unet of the latent diffusion. The input channel of the first convolution layer is set to 12. The training is conducted with a batch size of 64 and a learning rate of $1e-4$ over $4k$ steps. We use MagicBrush (Zhang et al., 2024) and InstPix2Pix (Brooks et al., 2023) as the training data in the second stage. $\alpha_I$ and $\alpha_T$ are set to be 1.5 and 7.5 respectively. All experiments are conducted in PyTorch on 2 80G A100 GPUs.

Please refer to the anonymous GitHub repo[2] for the implementation codes and collected benchmark CoReferEdit.

### E.1   INFERENCE EFFICIENCY

We compare with baselines InstPix2Pix (Brooks et al., 2023) with latent diffusion backbone and MGIE (Fu et al., 2023) with latent diffusion and LLM backbone in terms of inference efficiency. The time consumption are fairly compared with an A100 GPU of batch size of 1. On average, one turn of edit costs 4.46 sec, 9.82 sec and 7.86 sec for InstPix2Pix (Brooks et al., 2023), MGIE (Fu

---

[2]https://anonymous.4open.science/r/ReferPix2Pix

et al., 2023) and our model respectively, as shown in table 5 While both MGIE (Fu et al., 2023) and our approach employ MLLM for latent diffusion editing guidance, our method requires only a single [SEG] token for pixel-grounded guidance, in contrast to MGIE (Fu et al., 2023) that needs to generate 8 visual tokens. This efficiency enhances our model's inference speed over MGIE (Fu et al., 2023).

| Method | # Trainable Params | Inference time (s/img) | FLOPs (T) |
|---|---|---|---|
| InstructPix2Pix | 1.1B | 4.46 | 0.76 |
| MGIE | 2.0B | 9.82 | 3.55 |
| Ours | 1.3B | 7.86 | 1.87 |

Table 5: Our model produces a single [SEG] token for editing guidance, whereas MGIE requires multiple token generation.

