# OpenReview forum: "ReferPix2Pix: Guiding  Multi-Modal LLMs for Image Editing with Referential Pixel Grounding"
_ICLR.cc/2025/Conference — ICLR 2025 Conference Withdrawn Submission_

### Official Review · Reviewer_Wymt · 2024-10-17

**Soundness:** 3
**Presentation:** 2
**Contribution:** 2
**Rating:** 5
**Confidence:** 3

**Summary:**

This paper presents a novel image manipulation paradigm designed for user-friendly instruction and enhanced controllability. It introduces Multimodal Language-Driven Models (MLLMs) that interpret editing instructions and identify regions of interest. Additionally, the CoReferEdit benchmark is proposed to effectively evaluate editing capabilities.

**Strengths:**

1. From a technological standpoint, this paper extends the MGIE framework by integrating SAM to effectively address specific instructions related to visual grounding. The approach of generating spatial guidance through Multimodal Language-Driven Models (MLLMs) is both interesting and intuitive.

2. This paper highlights the limitations of current instruction-based image editing methods, particularly their difficulty in spatially modifying specific objects.

3. An LLM agent is introduced to clarify desired editing operations by providing precise editing masks or refining instructions. This approach leverages the LLM's strong planning capabilities, enhancing model comprehension in an intuitive manner.

**Weaknesses:**

1. A primary concern is whether referring expressions truly offer a user-friendly approach. If users aim to edit a specific region, would directly drawing a mask be more convenient? This raises questions about the task's formulation, which appears to combine visual grounding and image editing.

2. Another technical concern is the necessity of employing a Multimodal Large Language Model. For example, can the intended region be identified using SAM and a standard LLM with appropriate prompting? The scope of these specific instructions seems narrower than that of MGIE. Could a standard LLM rather than MLLM achieve comparable results?

3. Regarding fair comparison with MGIE, is it possible that MGIE could achieve similar performance with careful setup? For instance, with slight adaptations, MGIE might generate prompts that yield similar masks using the SAM model.

4. The overall presentation quality could be improved. For instance, in Figure 2, the input is positioned in the center while the output is in the upper-left corner, which may not align with conventional reading habits.

**Questions:**

See the weakness section.

---

### Official Review · Reviewer_VS9x · 2024-11-04

**Soundness:** 2
**Presentation:** 4
**Contribution:** 2
**Rating:** 5
**Confidence:** 5

**Summary:**

This paper studies instruction-guided image editing. Specifically, the paper aims to solve the instructions with a reference, which is challenging. To solve this task, it proposes ReferPix2Pix to train a MLLM to predict the mask of the referred object, and use it as another input to the diffusion model. Evaluations show that the proposed method outperforms baselines in various tasks.

**Strengths:**

- The paper is well written. Various nice tables and figures visualize the method and comparison results.
- Extensive implementation details are provided, e.g., editing instructions, along with the code. These significantly help the reproducibility.
- Compared with the baseline IP2P and MGIE, the proposed method performs well in many editing tasks.
- A novel benchmark, "CoReferEdit," is provided for the task.

**Weaknesses:**

- Some of the related works have not been investigated thoroughly or accurately.
   - At L124, "SDEdit" (Meng et al., 2021) was claimed to be "controlling cross-modal attention maps between global description and latent pixels." However, this mismatches with the method of SDEdit - its idea is to control the adding noise and denoising steps, while there is even no occurrence of the word "attention" in its paper.
   - An important branch of methods are not discussed: inversion-based methods, e.g., DDIM-inverse [Null-text Inversion for Editing Real Images using Guided Diffusion Models] and DDPM-inverse [An Edit Friendly DDPM Noise Space: Inversion and Manipulations]. Even though these methods need to input the description of the edited image instead of the instruction, the conversion between them can be easily done with any LLMs. Therefore, these methods should also have been compared.
- The method is quite similar to another earlier (but also not cited) paper, and shares similar disadvantages: InstructEdit [InstructEdit: Improving Automatic Masks for Diffusion-based Image Editing With User Instructions].
   - "InstructEdit" also uses a specific model to support reference masking but used DiffEdit to directly inpaint the masked part to fit the new prompt. ReferPix2Pix, though, did not use such a direct way to deal with the mask, but still highly guided the results with the mask.
   - As a masked-dependent method, it may share the common disadvantages: The mask strictly constrains the maximum region of the edited contents to always be within the mask, which is the original object. Therefore, these methods might be intrinsically challenging when editing a small object into a large one or adding lots of objects in a complicated frame.
    - The editing tasks in this paper look quite patterned and seem ignoring the
    - This disadvantage is observed in Fig.7, where "add him a hat" in the proposed method seems just to color a part of the head to the color of a hat in order to make it look like wearing a hat.
    - Therefore, I would like to see more about these results, e.g., add a large cowboy hat, turn a small cat into a giant tiger, add a person in front of a complicated room, etc. I would also like to see some object moving or motion-related editing tasks, like "make the person's arm raise up."
- (Minor) The idea of training MLLM/VLM to deal with text and image relationships is a standard approach and is very common.

**Questions:**

Please refer to "Weaknesses", especially the more challenging editing tasks that might fail in masked-related methods.

---

### Official Review · Reviewer_q8M5 · 2024-11-04

**Soundness:** 3
**Presentation:** 3
**Contribution:** 2
**Rating:** 5
**Confidence:** 4

**Summary:**

This paper proposes the use of Multi-Modal Large Language Models (MLLMs) for image editing, with particular emphasis on referring expression comprehension and multimodal co-reference in interactive editing tasks. The aim is to facilitate more natural editing aligned with user commands. The authors bridge MLLMs and Segment Anything Model (SAM) with a [SEG] token, fine-tune the MLLM, and enable the [SEG] token to be decoded by the SAM decoder. Additionally, they construct a CoReferEdit benchmark to evaluate co-reference editing capabilities.

**Strengths:**

1. The authors' novel approach of combining MLLMs and SAM with a [SEG] token is innovative and intriguing.
2. The use of state-of-the-art Vision-Language Large Models (VLLMs) such as GPT-4V to construct training and evaluation data demonstrates.
3. The establishment of the CoReferEdit benchmark to evaluate co-reference editing ability addresses a gap in previous benchmarks.

**Weaknesses:**

1. The authors' description of the SAM input as encoded image features and projected last hidden h[seg] lacks clarity. Further explanation of the rationale behind this design would be beneficial. It would also be informative to know whether different choices of h[seg] affect performance and if any ablation studies were conducted.
2. There is a potential concern regarding overfitting, as the editing assistant consistently responds with "Sure, it is [SEG]." This warrants further investigation and discussion.
3. Some annotations in the paper are not fully explained. For instance, the definitions of L_caption and L_mask are not provided.
4. Figure 2 appears to lack essential annotations, captions, and explanations. It is recommended that the authors align the figure with the formulas presented in the paper to enhance clarity and coherence.

**Questions:**

see weakness

---

### Official Review · Reviewer_knpq · 2024-11-07

**Soundness:** 3
**Presentation:** 2
**Contribution:** 2
**Rating:** 5
**Confidence:** 4

**Summary:**

This paper introduces ReferPix2Pix, an approach for text-guided image editing using multimodal large language models (MLLMs) to improve accuracy when editing specific instances within complex images. The paper is well written and approach is simple to follow. The model is effective for editing images with multiple objects, leveraging pixel-level guidance from MLLM to accurately interpret and execute instructions. The authors also present a new benchmark dataset, CoReferEdit, to evaluate the model's performance in resolving co-references across iterative editing rounds. The authors present quantitative and qualitative results on different benchmarks which proves the efficacy of their approach. The contribution of the paper beyond the dataset is highly incremental as the results are quite similar to other approaches such as InstructPix2Pix on datasets such as MagicBrush, while the other methods are not adapted to referring expressions. The paper would also benefit from discussion on how multi-turn edits affect the diffusion model , as details of what editing instructions are passed are unclear (local caption vs global caption vs edits only). The paper is at borderline for the high bar for ICLR.

**Strengths:**

1. New Benchmark: The CoReferEdit benchmark offers a standardized way to assess models on iterative co-reference resolution in image editing, filling a gap in current evaluation datasets, given novel use of existing datasets, which will benefit the community
2. Enhanced Precision: The model offers accurate editing of specific instances in images with multiple similar objects by integrating pixel-based guidance.
3. Effective Co-reference Resolution: ReferPix2Pix can resolve ambiguous instructions across multiple editing rounds, making it highly effective for iterative editing.
4. Strong Empirical Results: The model outperforms existing methods like InstPix2Pix, HIVE, and MGIE in various metrics (e.g., CLIP similarity and DINO score), showcasing its superior capability in complex editing tasks.

**Weaknesses:**

1. The approach is not novel as it utilizes a mask based approach and is specific to coreference dataset to edit the diffusion model, with segmentation mask doing the heavy lifting.
2. The results shown for Magicbrush dont match the results shown in the Magicbrush and MGIE paper, clarification in differences to the cited papers will help clarify impact.
3. The  implementation details are missing, such as MLLM being trained is not mentioned, and the details of the diffusion model which is trained are missing.
4. Potential Overfitting to Complex Scenarios: The model is designed for multi-instance editing scenarios and does not demonstrate the same level of improvement in simpler tasks.
5. Missing intuition to mask based editing baselines such as AnyDoor [CVPR 2024] [https://arxiv.org/pdf/2307.09481]

**Questions:**

1. Please share details of the MLLM being trained and how does it generalize to edits beyond the dataset, specifically in multiturn scenarios with conflicting instructions, such as addition and removal of an object in single session of interactive editing.
2. Intuition behind on how the model would perform if the referred object is switched during the instruction editing
3. Discussion on how the objects and classes chosen for the dataset creation, were the objects and classes paired by LLM. Did the edits have any nonsensical edits ?

---

### Note · Authors · 2024-11-14

I have read and agree with the venue's withdrawal policy on behalf of myself and my co-authors.